

**Spatial relationship between hydrodynamic and physico-chemical parameters of surface**
**water for a basin with shale rock series as an indicator of intensity and direction of**
**chemical denudation in the Western Carpathians**
Edyta Kruk[1], Wiktor Halecki[2], Marek Ryczek[3], Agnieszka Petryk[4],
Krzysztof Chmielowski[5], Paweł Guzdek[6]
[1]Department of Land Reclamation and Environmental Development, Agriculture University of Krakow,
Mickiewicza 21 St, 31-120 Krakow, 30-059 Krakow, Poland; phone: +48 12 662 40 15, edyta.kruk@urk.edu.pl
[2]Institute of Nature Conservation Polish Academy of Sciences, Mickiewicza 33 St., 31-120 Kraków, Poland, phone:
+48 12 632 22 21, e-mail: halecki@iop.krakow.pl
[3]Department of Land Reclamation and Environmental Development, Agriculture University of Krakow,
Mickiewicza 21 St, 31-120 Krakow, Poland; phone: +48 12 662 40 15, marek.ryczek@urk.edu.pl
[4]Correspond author Department of Space Management and Social-Economic Geography, Krakow University of
Economic , Rakowicka 27 St., 31-510 Kraków, Poland, phone: +48 12 293 74 20, e-mail: petryka@uek.krakow.pl
[5]Department of Natural Gas Engineering, AGH University of Science and Technology, Mickiewicza 30 St., 30-059
Krakow, Poland; phone: +48 12 617 31 53, krzysztof.chmielowski@agh.edu.pl
[6]Cracow University of Technology, Department of Water Supply, Sewerage and Environmental Monitoring,
Warszawska24 St., 31-155 Kraków, +48 12 6282 83, pawel.guzdek@pk.edu.pl
**Keywords:**
hydrodynamic parameters, physico-chemical parameters of water, indicator of intensity, chemical
denudation
**Abstract**
Hydrochemical evaluation of stream quality in the Western Carpathians requires a system
approach, gradually excluding factors less or more responsible for washing, mixing contaminants
and their farther transportation in the stream channel. In this work, the spatial autoregression
model was used to estimate the relationship between hydrodynamic and physico-chemical
parameters of surface water in various groups and variants of basin use. The highest mean shear
stress was 0.178 N·m$^2$ in forest surface water. The highest mean Reynolds number (23654) was
recorded in the stream channel at permanent grassland, and the lowest number (0.426) at arable
lands.Analysis of spatial autoregression to a high degree showed space-time relations in various
measurement points. The turbulent diffusion coefficient should be regarded in the space-physical
model, constructed based on the influence of hydrodynamic indicators on the shaping of physico-
chemical parameters in the flysch basin. The autoregression confirmed that the turbulent
diffusion coefficient played a high role for ions $K^+$ and $P\text{-}PO_4^{3-}$ in surface water at arable lands



and for cation $K^+$, as well as total iron for grassland ($p<0.05$). A relation for physico-chemical
was not found for surface water in forests. The results, to a high degree, will be used to createan
erosion model concerning the alimentation of alluvial deposits from weathered Carpathian flysch
or surface wash depending on the material delivery in a basin.
**1.      Introduction**
The factors influencing the physico-chemical state of surface water to the highest degree
are soil, relief (Shi et al., 2016),and soil plant cover occurring in the river-bank zone (Andersson
et al., 2015; Teixeira and Marques, 2016). In flysch basins, surface runoff is selective because
transported material often undergoes local accumulation on a slope during delivery to a stream
(Gil and Kotarba, 1977).Reliability of results is ensured when the picture of surface erosion is
being captured in mountain valleys, and proper frequency of measurement series is carried out,
together with specification of measurement error (Halecki et al., 2018a).
Investigations of water erosion can be carried out from a theoretical point of view and the
use of mathematical models (Panagoset al., 2015), for example,in the aspect of modelling the
scape of nitrates in surface runoff (Wang et al., 2014). Further, empirical equations and
theoretical methods are used for the settlement of river material quantity in various measurement-
control points (Mazur and Pałys, 1992) and intensity degree of suspended sediment transportation
in the shape of mechanical weathered rock mantle (Izmaiłowet al. 2008; Starkel 2011, Bryndalet
al. 2014; Comino et al., 2016). It is important to remember that technical solutions simulating
digital sediment transportation are only approximate and do not reflect real data. The cause of
unreliable results is the short or irregular frequency of field measurements (Haleckiet al., 2018 b).
Material transported in the Beskidy region basins is connected with land use and the
occurrence of shale rock series (Haleckiet al., 2018 c). Material delivered from slopes determines
the concentration of particular ions transferred in the stream channel. Ions can originate from the
dissolution of mineral fertilisersfrom adjacent areas of agricultural use (Hall et al.,2014; Gernezet
al., 2015). The fundamental biogenic indicators in the hydro-chemical evaluation of surface water
are  phosphates  and  nitrates.  Nitrogen  most  often  appears  as  an  ammonium  ion
($N-NH_4^+$), testifying point contamination of surface water. Nitrate  ($N-NO_3^-$) and nitrite nitrogen
($N-NO_2^-$) are indicators of the long-term influence of pollutants, particularly in the proximity of
crops. In summer and early autumn, $N-NO_3^-$ decreases, which is connected with the plant's



demand for nutrients (Ulénet al., 2012). The concentration of calcium ($Ca^{2+}$), magnesium ($Mg^{2+}$),
sodium ($Na^+$), potassium ($K^+$), total iron ($Fe_{tot}$), and nitrite nitrogen ($N-NO_2$) is differentiated in
spring.Thisshows thatplants accumulate more nutrients during the warmer season(Padmalalet al.,
2012). For estimation of the physico-chemicalquality of surface water, the good indicators
regarding transformation and accumulation are various forms of phosphorus. They pose a high
threat to the purity of surface water as factors that favour eutrophication and excessive rise of
biomass of particular algae (Smoroń, 2012). Changes in biogen concentration in surface streams
are influenced by the spatial distribution of arable land (Arienzoet al., 2012; Tasdighiet al., 2017).

Sedimentary rock undergoes an intensive leaching process. In surface water, sodium

cations can originate from industrial waste and potassium from agricultural cultivations, where
potassium fertilisersare used (Oster et al., 2016). Hydrolytic decay of minerals containing sodium
salts (aluminosilicates) and weathering of sedimentary rocks contribute to the creation of
alimentation (source) for sodium and potassium cations in surface water (Zhang et al., 2017).
Magnesium salts occur in all natural waters, both surface and underground. Leached $Ca^{2+}$and
$Mg^{2+}$ cationspenetrate the basin by infiltrating rainfall from fertilisedcultivated areas and
supplying surface water (Grochowska, 2016). Increased concentration of $Ca^{2+}$and $Mg^{2+}$cations in
surface water testify to the occurrence of calcareous rocks, marls, and dolomites and depends on
the degree of basin management (Halecki, 2015). In addition, the concentration of $Ca^{2+}$cations
and $SO_4^{2-}$anions in small stream channels is attributed to geology, mainly the calcium carbonate
and sulphate concentration in the clastic rockmantle. Moreover, certain water quality factors
require specialistic investigation and hydrochemical evaluation according to the World Health
Organisation (WHO) standards. These include $Ca^{2+}$ and $Mg^{2+}$cations,$N-NO_2^-$, $N-NO_3^-$, $SO_4^{2-}$, and
phosphate phosphorus ($P-PO_4^{3-}$) anions, and salinity indicators - specificallythe electric
conductivity of water and the concentration of dissolved substances (sum of determined mineral
substances in a shape of fine clastic material) with total suspension (Shigutet al., 2017).

Dissolved oxygen is essential for the evaluation of water conditions. The activity of

microorganisms responsible for oxidation of organic compounds is examined using two
indicators, namely biochemical oxygen demand (BOD) and chemical oxygen demand (COD). In
evaluating surface water quality, higher oxygen use indicates contamination (Bo et al., 2017;
Effendi et al., 2018). On the other hand, a lower dissolved oxygen concentration can show
biochemical decay of accumulated organic substances, respiration of water organisms, or



oxidation of inorganic substances (Matta et al., 2017).Furthermore, water temperature controls the solubility of solid substances and the concentration of suspension (Saito et al., 2005a). So, low temperature weakens the chemical and biological activity of water and increases viscosity, enabling the transportation of material (Jarocki, 1957). Higher values of dissolved oxygen and BOD are connected with their high sensibility to changes in water caused by the activity of water engineering (Parmar and Keshari, 2012). Essential differences between BOD and COD of anthropogenic origin have been recorded in a seasonal hydro-chemical evaluation period (Bellver-Domingo and Hernández-Sancho, 2018). Seasonal evaluation of temperature, pH, COD, BOD and heavy metals: Fe, Mn, Ni, Cd, Cr, Co, Cu, Pb and Zn, is needed either for determination of the recommended level and permissible concentration of pollutants for drinking water (Vincent-Akpuandin, 2015). The anthropogenic activity contributes to delivering heavy metals to surface water, mainly by using industrial objects (Juahiret al., 2010; Weber et al., 2014; Assoulineet al., 2015; Hu et al., 2015; Vaddeet al., 2018).

Contamination of surface water is proven by the decay of living organisms, which mineral composition is related tothe increase of sulphates. Chlorine is also present in the form of chloride anions (Cl⁻). In surface water, it is transported as a result of weathering of rock-forming minerals. Dissolution of evaporates (sedimentary rock) enriched in anhydrite constitutes a natural source of $SO_4^{2-}$ anions. Further, sulphate ions do not undergo sorption in an underground environment, and similar to Cl⁻, it is the indicator of pollutant penetration to underground water (Sapek, 2008; Geurtset al., 2009).

The first detailed aim of this study was to arrange useful values of the basin and, secondly,to conduct a hydro-chemical evaluation of the surface water in various land configurations of the basin lithology. The third aim was to determinehydraulic conditions in the flysch stream channel. A final goal was to determine the main factors intensifying short-term relations between hydrodynamic parameters of mountain streams and chemical compounds leached from the slope to the water of the flysch stream as a result of interrill erosion. For such detailed aims, hypotheses were formulated, methods and schemes of frequency measurement series were set, and specifications of measurement errors were elaborated.

## 2. Methods



**2.1. Investigated site and uptake and examination of surface water samples**

The measurements were carried out in the Smugawka stream in the Beskid Wyspowy, the Western Carpathians (Poland). The localisationof sampling points and land use is presented in Figure 1. Slopes and exposition are shown in Figures 2. Water samples were taken in 1 dm$^3$ volume containers once a month to determinethe physico-chemical parameters of surface water. The sampling period occurred in the spring-summer season (from March) and fall-winter (to November) from 2014 to 2018. Sampling was carried out punctually, in cross-section levels, in source and outlet places for the dimension of samples to be multiplied. After filling, each container was emptied and then filled with flowing water for 120 seconds.

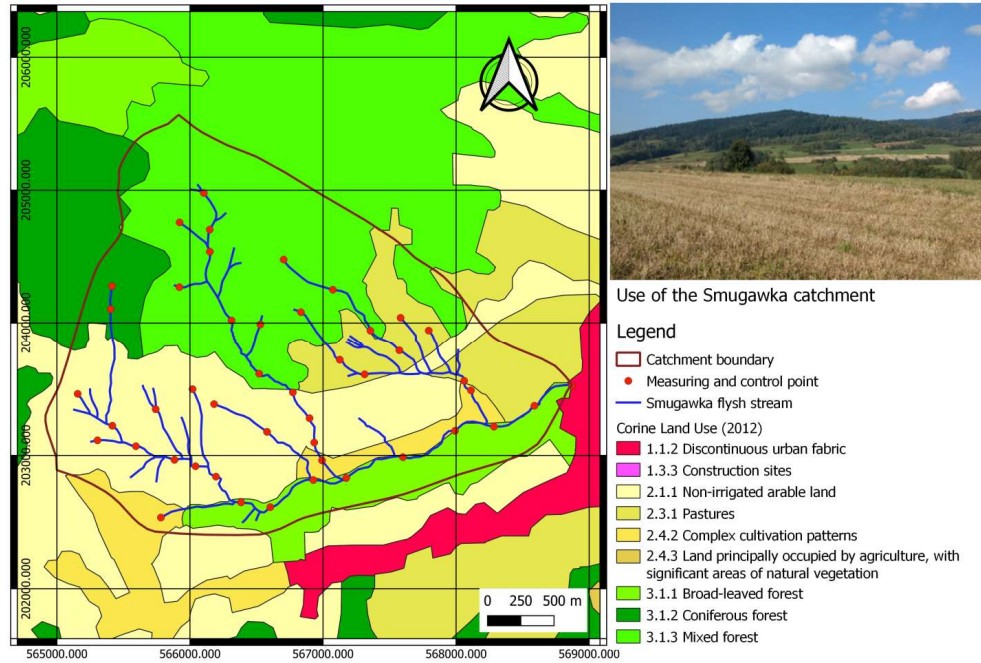

*Figure 1. Localisation of sampling points and land use*

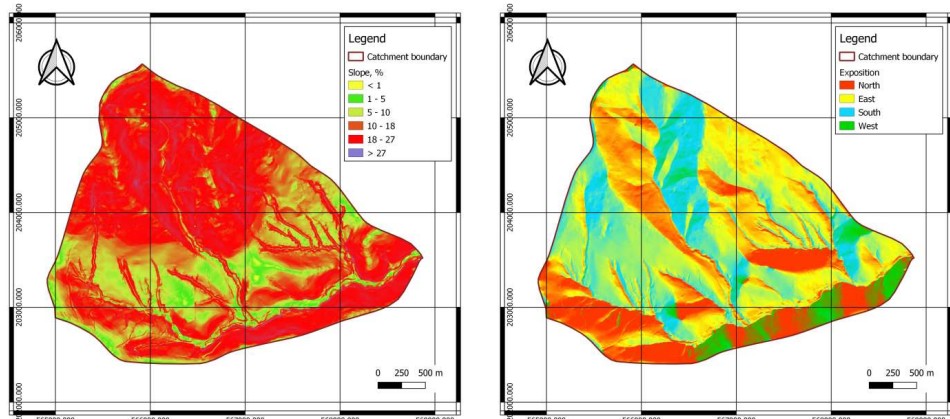

*Figure 2.* Slope and exposition of the Smugawka stream catchment
One water sample was enough to determine the mean concentration of dissolved material
(Brański, 1968; Dalbianco et al., 2017) because it represents all hydrometric sections of the
channel in which strong longitudinal and transversal dispersion occurs. Moreover, the
relationship between the concentration of dissolved material is directly proportional to water flow
and in accordance with the hydrological system of a stream (Wojtasik and Szatten, 2014). Six
samples were taken during each field visit. Measurement of convection intensity was carried out
with flow intensity. At <0.3 m, water was taken directly to the container. After determining the
relationship between the mean concentration of total suspension in the section and the
concentration of punctual sediment in a cross-section, the number of samples was limited to 2-4.
The database contained 24 variables (physico-chemical features) in 25 replications. In summary,
600 samples were takenfor laboratory analysis. To assure comparability of results, investigations
into physico-chemical surface water quality during field visits (additionally there were carried out
250 measurements) were conducted, recording actual results at low water levels and during
freshets.
**2.2. Hydro-chemical indicators**
Geodetic measurements were madeusingthe tachymeter TOPCON GTS-226. Five cross-
sections of the bed channel slopes, from 0.79 to 2.18%, were localisedalong all stream lengths
(from sources to outlet). The average distance between them was 875 m. Granular measurements
of bottom sediment were carried out using the Wolman method (Wolman, 1954). In every





measurement point, 15 cobbles were taken, and their mean axis „b" was measured. The grain size
distribution curves were drafted, and their characteristic diameters: $d_{min}$, $d_{30}$, $d_{50}$, $d_{70}$, and $d_{max}$ and
competent diameter: $d_m$ were determined. Measuring points were located crosswise channel in a
distance of 0.5 m.

Hydrodynamic measurements were carried out at the same points as the granular ones.

Measurements were made using a hydrometric current meter VALEPORT Model 801 Flat EM
flow meter. This device allows the measurement of mean flow velocity in assigned time intervals.
It also measures instantaneous velocity and filling in the stream channel. The measurement scope
of the device is 5.0 m·s$^{-1}$, and exactness ± 0.001 m·s$^{-1}$. Through this device, mean velocity $V_{mean}$ on
the height 0.4 m above the bottom, maximum velocity $V_{max}$, and instantaneous velocity $V$
measured just above the bottom, allowed calculation of dynamic velocity based on charts of
velocity distribution above the bottom in a semi-logarithmic system (Gordon et al., 2007):

$$V_* = \frac{a}{5.75}\,[m \cdot s^{-1}]$$

where:
a = coefficient of straight line slope
V = f(h), taking an equation form of y = ax + b (x = height above bottom, the measurement of
velocity was carried out, b = free term of the equation).

Knowing dynamic velocity, shear stress was calculated as follows:

$$\tau = \rho \cdot (V_*)^2\,[N \cdot m^{-2}]$$

where:
$\rho$ = water density (kg · m$^{-3}$)

The Reynolds number (mean, maximum, and grain) was used to determine the type of

flow pattern as laminar or turbulent. Further, the Freud number (mean and maximum) is a
measurement of bulk flow characteristics. These were respectfully calculated using the following
equations:



$$Re_{śr} = \frac{V_{śr} \cdot h}{\upsilon}$$

$$Re_{max} = \frac{V_{max} \cdot h}{\upsilon}$$

$$Re_{dm} = \frac{V_* \cdot d_m}{\upsilon}$$

$$Fr_{śr} = \frac{V_{śr}}{\sqrt{gh}}$$

$$Fr_{max} = \frac{V_{max}}{\sqrt{gh}}$$

where:
h = filling (m),
g = acceleration of gravity (m·s⁻²),
$\upsilon$ = kinematic coefficient of water viscosity, calculated from the equation:

$$\upsilon = \frac{0.00178}{\rho(1 + 0.0337t + 0.000221t^2)} [m^2 \cdot s^{-1}]$$

where:
t = water temperature(0°C)

In this study, hydrodynamic indicators allowed the calculation of the distance of full mix
in surface water, used in the analysis of spatial autoregression. Dynamic velocity (u) in the stream
was calculated using the following equation (Loga, 2016) :

$$u = \sqrt{h \cdot g \cdot S} \ [m·s^{-1}],$$

where:
h = mean height of water (m),
g = acceleration of gravity (m·s⁻²),
S = slope of channel bottom (%).

There is a risk connected to the negative influence of the frequency of measurements.
In short periods of freshets, some part of the yearly sediment load cannot be observed. Therefore,



the following equation was used to present a moment in which the mix of material in water
occurs (a load of total suspended sediment and physico-chemical composition). That is some
quantitative representativeness of transported material. In addition, the turbulent diffusion
coefficient ($D_{t,y}$) was calculated for transverse mix in the particular years in a so-called natural
channel (Loga 2016):

$$D_{t,y} = 0,6 \, h \cdot u \quad [m^2 \cdot s^{-1}],$$

where:
h = mean height (m),
u = dynamic velocity ($m \cdot s^{-1}$).

**2.3. Determination of physico-chemical properties of surface water**

In the experimental area,the physico-chemical properties of surface water were measured.
The reaction pH of water was examined usingthe potentiometric method with a CP-104
ELMETRON equipped with a combined electrode. Specific electrical conductivity, expressed in
$dS \cdot m^{-1}$, was measured with a conductometer Elmetron CC-101. The concentration of dissolved
oxygen in ($mg \cdot dm^{-3}$) and the degree of water saturation by oxygen were determined using the
electrochemical method usingan oxygen-meter Elmetron CO-411. Further, the water temperature
(°C) was measured by a digital thermometer built in waterproof oxygen-meter CO-411. The total
content of dissolved substances in ppm was calculated by means of a digital TDS device. In the
laboratory,the following physico-chemical properties of water were determined using the
UNICAM SOLAAR 969 atomic absorption spectrophotometer: $Ca^{2+}$, $Na^+$, $K^+$, $Mg^{2+}$, $Fe_{tot}$, and
manganese $Mn^{2+}$.
Moreover, $N-NH_4^+$, $N-NO_3^-$, $N-NO_2^-$, $P-PO_4^{3-}$, and $Cl^-$ were determined by the colorimetric
flow-injection method on the computer-controlled FIA Star 5000 apparatusof the FOSS
firm.Sulphates ($SO_4^{2-}$) were determined gravimetrically. Biological Oxygen Demand($BOD_5$) was
analysed by the Winkler method: water sampleswere treated with manganese sulfate $MnSO_4$,
alkaline potassium iodide KI, and sulfuric acid $H_2SO_4$, followed by titration with sodium
thiosulfate $Na_2S_2O_3$. The chemical oxygen demand (ChZT-Mn) was calculated by the



permanganate method by heating the water sample with $KMnO_4$ potassium permanganate in an
acidic environment, and the amount of consumed oxygen was determined by titration.
During subsequent readings of water levels, samples were taken from the stream by the
bathometric method (containers with a volume of 1 $dm^3$). The concentration of total suspended
solidswas determined by the gravimetric method after drying using tared filters with an accuracy
of ± 0.0005 g. The dry residue (sum of mineral particles from the transported samples) from the
collected material was filtered to determine the different composite concentrations. The ionic
forms of zinc ($Zn^{2+}$), lead ($Pb^{2+}$), cadmium ($Cd^{2+}$) and copper ($Cu^{2+}$) were determined once per
quarter with the coulometric method using an electrochemical analyser for the determination of
trace amounts of heavy metals (EcaFlow 150 GLP by PolEko). The nephelometric-laser method
was also used to assess the degree of water turbidity using a Hach Lange 2100QS nephelometer
(turbidimeter) in the range of 0–2000 FNU, per the international standard (Nephelometric
Turbidity Unit; NTU).

### 239 2.4. Data reduction to a spatial model and statistical analysis

Before the analyses were performed, the asymptotic distribution for the $\chi^2$ Jarque-Ber (JB)
test statistic was studied, which takes kurtosis and skewness into account (Jarque and Bera,
1987). The data was grouped in ascending order by the Anderson-Darling concordance test,
determined by the empirical distribution, and the normalisation for the mean and standard
deviation was calculated (Stephens, 1986). The test determines a weighted Cramér-von Mises
distance between the empirical (Fn) and theoretical (F) cumulative distribution factors, with
weights corresponding to the reciprocal of the empirical cumulative distribution (Anderson and
Darling, 1954). The Shapiro-Wilk test, based on positional statistics, was also used. The keynote
of this analysis is collinearity between the empirical quantile (i.e., the ordinal statistic) and the
theoretical quantile (i.e., the expected value of the ordinal statistic) along the y = x straight line
(Shapiro and Wilk, 1965).
The homogeneity of variance was analysed by Levene'stest (Levene, 1960). To reduce the
number of variables, the Principal Component Analysis (PCA) was used, presenting the
relationships between significant physico-chemical properties in the surface water of the stream
running near arable land, grasslands, and forests. Statistical significance for the matrix of
correlation coefficients and the strength of the relationship between the variables were assessed



with the Barlett sphericity test (Bartlett, 1950; Williams et al., 2010). Partial correlations with
bivariate correlation coefficients were performed using the Keiser-Mayer-Olkin coefficient
(KMO), determining the accuracy of selecting variables for the tested model. Adequate variables
have a KMO coefficient> 0.5 (MacCallum, 1983; Hair et al., 1995; Tabachnick and Fidell, 2007;
Szüle 2016).
In order to show the relationship between the tested physico-chemical properties of
surface water, values for eigenvectors were given, approximating the influence of primary
variables on the main component. The most significant variables were interpreted through factor
loadings, which also reflect the influence of individual variables on a given principal
component.The study useda correlation matrix, and the factor loadings were interpreted as
correlation coefficients between the original variables and the next analysed main component.
Following the PCA method, the variance size for each computed component and the so-called
primary variableswere indicated. The most important variables were selected for further data
processing based on the value of factor loadings. The analysis was performed in PQ Stat
Software, version 1.6.6.
Spatial autoregression is a tool for predicting features in a linear system. The analysis
aimed to determine the factors that influence the temporal changes in the physico-
chemicalcomposition ofsurface water. Therefore, the study used a spatial autoregression model
with a lagged response, facilitating the study of the relationship between the distance of complete
mixing and physico-chemical features randomly distributed in the water column. The general
form of spatial autoregression is given by the formula (Rangel et al., 2010):

$$Y = \rho \cdot Wy + X\beta + \varepsilon$$

where:
$Y$ = vector ($N \cdot 1$) of the explained spatial process through a responsive (explained) variable,
$W_y$ = weight matrix of the vicinity of the examined variables in different locations ($N \bullet N$ matrix
of spatial weights),
$\rho$ = spatial autoregression parameter reflecting the strength of the relationship between the
variables (spatial interaction parameter),
$X$ = matrix ($N \cdot K$) of explanatory processes,
$\beta$ = vector ($K \cdot 1$) of structural indices,





ε = vector (N · 1) of the random term (estimation error and spatial noise not described by the
model; random effect).

The spatial autoregression model was usedto show the relationship between the variables

distributed in space over the courseof a measurement series for homogeneous areas and to
facilitate the analysis of the variables, especially with water flows greater than average. The
transverse turbulence diffusion coefficient was assumed as the dependent variable, and surface
water's physico-chemical features were considered independent variables. The simultaneous
autoregressive model (SAR), with a system of lagged predictors (Rangel et al., 2010), was used
to calculate the spatial dependencies:

$$C \;=\; \sigma^2 [\,(\,I - \rho \cdot W\,)^T\,)^{-1}[\,I - \rho \cdot W\,)^{-1}$$

where:
C = vector of the explained spatial process through a responsive (dependent) variable,
W = weight matrix of the vicinity of the examined variables in different locations matrix (N • N)
of spatial weights,
$\sigma^2$ = residual variance between observations,
ρ = spatial autoregression parameter reflecting the strength of the relationship between the
variables (spatial interaction parameter),
I = linear transformation (matrix of N ·N-type) of the dependent variable.

The autoregression analysis documented the overarching variables for each type of land

use, combining the physico-chemical characteristics of the surface water. The calculations were
made in the SAM 1.6.6 software.







## 3. Results

### 3.1. Physico-chemical and hydrodynamic features of the stream bed

The range of concentrations of the salinity indicators is presented in Table 1, and the range of concentrations of metals, including heavy metals, for the Smugawka stream throughout the study period is provided in Table 2. In surface water flowing out of arable land, a strong positive correlation was found for $Ca^{2+}$ and $P\text{-}PO_4^{3-}$ for the first two factors of principal component analysis (Table 3). In grassland, the water temperature played an important role, distinguishing strong factor loadings. In contrast, the concentration of $Fe_{og}$ showed a positive correlation with the first and a high negative correlation with the second factor of the PCA (Table 4). Water temperature and concentration of ChZT-Mn were also characterised by the highest variance for the factor axes in all tested surface water samples and the highest factor loadings (Table 5 and Figure 3). The transport intensity increased with the following gradient: forests > grassland > arable land (Table 6).

Table 1. The range of concentrations of salinity indicators for the Smugawka stream during the study period

| Salinity indicators | Unit | Measuring point | | |
| --- | --- | --- | --- | --- |
| | | Surface waterrunoff from arable land | Surface water runoff from grassland | Surface waterrunoff from forests |
| $Na^+$ | [mg·dm$^{-3}$] | 4,80–88,31 | 3,91–68,72 | 8,92–37,54 |
| $K^+$ | | 1,78–5,20 | 0,68–10,40 | 3,20–5,40 |
| $Mg^{2+}$ | | 6,64–10,92 | 3,74–19,35 | 10,34–22,59 |
| $Ca^{2+}$ | | 39,53–133,08 | 31,22–147,23 | 58,63–80,82 |
| $SO_4^{2-}$ | | 43,87–204,24 | 23.56–193,36 | 45.05–120,93 |
| $Cl^-$ | | 23,94–134,45 | 35,05–195,36 | 24,59–145,48 |
| *CZSR | | 101–147 | 78–199 | 148–178 |
| Electricalconductivity | [µS·cm$^{-1}$] | 156–240 | 247–350 | 148–256 |

*CZSR – Total content of dissolved substances

Table 2. The range of concentrations of metals, including heavy metals, for the Smugawka stream during the study period

| Indicators of metal concentrations | Unit | Measuring point | | |
| --- | --- | --- | --- | --- |
| | | Surface waterrunoff from arable land | Surface water runoff from grassland | Surface waterrunoff from forests |
| $Zn^{2+}$ | [µg·dm$^{-3}$] | 12,53–50,10 | 2,93–41,90 | 19,94–47,80 |
| $Pb^{2+}$ | | 1,61–6,80 | 0,01–7,60 | 0,01–1,34 |
| $Cd^{2+}$ | | 0,01–1,10 | 0,01–1,90 | 0,01–1,50 |
| $Cu^{2+}$ | | 1,51–3,04 | 0,91–1,50 | 0,01–0,70 |





| Fe_og | [mg·dm⁻³] | 0,53–0,91 | 0,03–1,04 | 0,1–0,81 |
|---|---|---|---|---|
| Mn²⁺ | | 0,12–0,18 | 0,06–0,09 | 0,09–0,10 |

Table 3. Factor loadings obtained from principal component analysis (PCA) for the studied
physicochemical characteristics from surface water samples taken at arable land

| Variable | Factor 1 | Factor 2 | Factor 3 | Factor 4 | Factor 5 | Factor 6 |
|---|---|---|---|---|---|---|
| K⁺ | -0,72 | 0,60 | -0,10 | 0,05 | -0,06 | 0,46 |
| pH | -0,13 | 0,35 | -0,78 | -0,06 | 0,49 | -0,11 |
| N-NO₂ | 0,31 | 0,17 | 0,19 | 0,90 | 0,15 | 0,01 |
| Ca²⁺ | 0,88 | 0,92 | -0,03 | -0,07 | -0,01 | 0,14 |
| Mg²⁺ | 0,84 | -0,86 | 0,02 | -0,16 | 0,07 | 0,13 |
| P-PO₄³⁻ | 0,93 | 0,87 | -0,01 | -0,15 | 0,04 | 0,18 |
| Na⁺ | 0,87 | -0,78 | -0,31 | 0,13 | 0,19 | 0,23 |
| Total suspendedsolids | -0,15 | 0,24 | 0,78 | -0,21 | 0,52 | 0,04 |

Table 4. Factor loadings obtained from principal component analysis (PCA) for the studies
characteristicsphysicochemical characteristics from surface water samples taken in grasslands

| Variable | Factor 1 | Factor 2 | Factor 3 | Factor 4 | Factor 5 | Factor 6 |
|---|---|---|---|---|---|---|
| Temperature of water | 0,82 | 0,76 | -0,60 | 0,10 | -0,10 | 0,12 |
| Total suspendedsolids | -0,63 | 0,58 | -0,62 | -0,17 | 0,08 | -0,14 |
| Fe_og. | 0,75 | -0,63 | -0,15 | -0,22 | 0,22 | 0,19 |
| K⁺ | -0,59 | 0,51 | -0,19 | -0,53 | -0,20 | -0,13 |
| Ca²⁺ | -0,61 | 0,54 | 0,17 | -0,26 | 0,43 | 0,001 |
| Na⁺ | -0,85 | -0,73 | 0,04 | -0,03 | -0,13 | 0,11 |

Table 5. Factor loadings obtained from principal component analysis (PCA) for the studied
physicochemical characteristics from surface water samples taken in forests

| Variable | Factor 1 | Factor 2 | Factor 3 | Factor 4 | Factor 5 | Factor 6 |
|---|---|---|---|---|---|---|
| Temperature of water | 0,84 | 0,86 | -0,28 | 0,41 | -0,03 | 0,32 |
| ChZT-Mn | 0,92 | 0,87 | -0,51 | 0,21 | 0,33 | 0,08 |
| Na⁺ | 0,97 | -0,78 | -0,48 | -0,44 | 0,12 | 0,23 |
| Conductance | 0,23 | -0,35 | -0,30 | 0,21 | 0,53 | -0,59 |
| BZT₅ | -0,16 | -0,53 | 0,08 | -0,64 | 0,46 | 0,22 |
| Mg²⁺ | -0,38 | 0,13 | 0,55 | 0,13 | 0,50 | -0,07 |
| Dissolvedoxygen | -0,56 | -0,81 | 0,15 | 0,35 | 0,33 | 0,65 |
| Fe_og. | -0,51 | 0,84 | -0,13 | -0,01 | -0,17 | 0,10 |





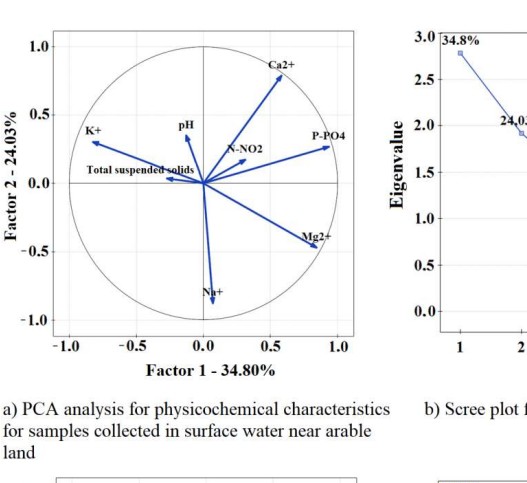

a) PCA analysis for physicochemical characteristics for samples collected in surface water near arable land

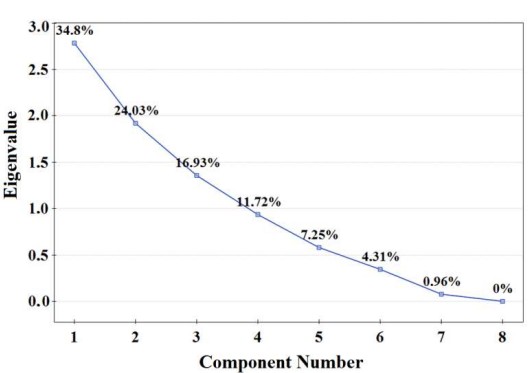

b) Scree plot for samples taken in surface water near arable land

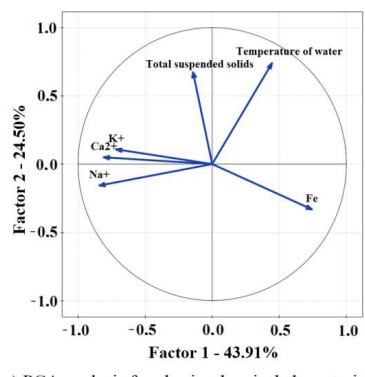

c) PCA analysis for physicochemical characteristics for samples collected in surface water from grasslands

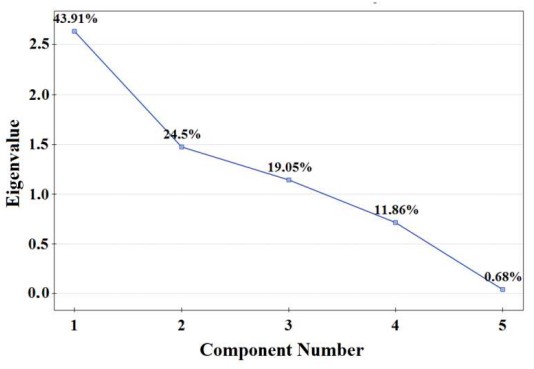

d) Scree plot for sample taken in surface water from grasslands

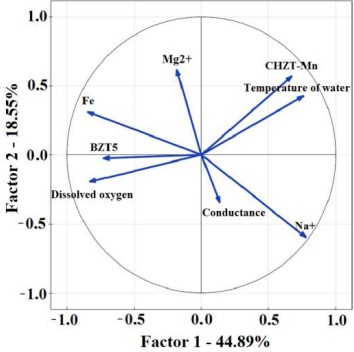

e) PCA analysis for physicochemical characteristics for samples collected in surface water near forests

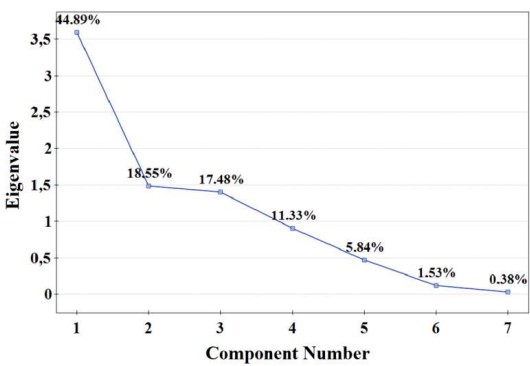

f) Scree plot for samples taken in surface water near forests

Figure 3. PCA analysis and scree plots for physicochemical characteristics for samples taken in surface water: arable
land (a, b), grassland (c, d), forest (e, f)



Table 6. Range of average values of soil material loss in the form of total suspended solids throughout the study period depending on the land use of the Smugawka catchment area

| Index of erodedsoilmaterial | Unit | Arable land | Grasslands | Forests |
|---|---|---|---|---|
| Circadianconcentration of totalsuspendedsolids | [mg·dm$^{-3}$] | 4,94–18,34 | 0,93–9,92 | 0,71–5,92 |
| Total suspendedsolidsload | [Mg] | 0,05–0,52 | 0,05–0,34 | 0,02–0,23 |
| Annualtotalsuspendedsolidsload | [Mg·rok$^{-1}$] | 18,69–190,27 | 19,65–126,95 | 7,38–53,78 |

### 3.2. Data reduction and spatial model

Spatial autoregression was used to reveal the relationships between the studied variables. Relationships with some physico-chemical features of surface water were detected in the model. The SAR equation for the entire catchment area showed that $P-PO_4^{3-}$, $N-NO_2^-$, TDS, and the concentrations $Na^+$, $Mg^{2+}$, $Ca^{2+}$ and $Mn^{2+}$ (Table7) are appropriate predictors for judging the distance of complete mixing. In general, the autoregressive modelling can be summed up by the fact that the few relationships for the physico-chemicalcharacteristics of the surface water with the transverse turbulence diffusion coefficient of the flysch stream were significant ($p < 0.05$). In the spatial autoregression model divided into variants of use, statistically significant relationships were found for $K^+$ and $P-PO_4^3$ in surface water ofarable land and for $Fe_{og}$ in surface water at grassland. No significant relationships were found at the measurement points in surface water near forests (Table 8). In the spatial autoregression model for the whole with the explanatory variable (the distance of complete mixing)detected the spatial dependencies for $P-PO_4^{3-}$, $N-NO_2^-$, $Mn^{2+}$, TDS, $Na^+$, $Mg^{2+}$, $Ca^{2+}$, and $Mn^{2+}$ (Table 9).

Table 7. The results of the spatialautoregression model of the SAR type for the entirecatchment. The - distance of fullmixing was chosen as the explanatoryvariable, and physicochemicalcharacteristics of surfacewaterweretaken as predictors of the model

| Physicochemicalcharacteristics | OLS ratio | SAR ratio | Standard error |
|---|---|---|---|
| Physicalcharacteristics | | | |
| Total suspendedsolids | -1,22 | 0,36 | 0,67 |
| Temperature | 10,41 | 16,28 | 5,90 |
| Oxygenindicators | | | |
| Oxygenconcentration | -0,46 | -0,23 | 0,19 |
| Degree of oxygensaturation | -0,03 | -0,04 | 0,01 |
| BZT$_5$ | -1,19 | -1,77 | 0,37 |
| ChZT-Mn | 6,46 | 4,55 | 1,09 |
| Biogenicindicators | | | |
| $P-PO_4^{3-}$ | 12,95 | 11,72 | 1,47 |


| | | | |
|---|---|---|---|
| N-NH$_4^+$ | 11,40 | 16,2 | 1,78 |
| N-NO$_3^-$ | -0,03 | 4,88 | 0,36 |
| N-NO$_2^-$ | 0,64 | 14,83 | 2,71 |
| Salinityindicators | | | |
| *CZSR | 5,70 | 8,53 | 0,27 |
| SO$_4^{2-}$ | -0,31 | 0,18 | 0,14 |
| Cl$^-$ | 0,06 | 0, 71 | 0,02 |
| Na$^+$ | 0,49 | -0,61 | 0,17 |
| K$^+$ | 2,92 | 10,95 | 1,06 |
| Mg$^{2+}$ | -0,46 | 6,32 | 0,82 |
| Ca$^{2+}$ | -2,93 | -4,41 | 1,41 |
| Electricalconductivity | -1,19 | 4,83 | -1,72 |
| Metals, including heavy metals | | | |
| Zn$^{2+}$ | -0,14 | 0,14 | 0,02 |
| Pb$^{2+}$ | -0,62 | 3,12 | 0,09 |
| Cd$^{2+}$ | -0,03 | 4,89 | 0,83 |
| Cu$^{2+}$ | 0,98 | 3,12 | 0,56 |
| Fe$_{og}$ | 6,46 | 12,81 | 1, 37 |
| Mn$^{2+}$ | 17,46 | 11,27 | 1,45 |

*CZSR – Total content of dissolvedsubstances

Table 8. Results of the spatial autoregression model of the SAR type (The transverse turbulencje
diffusion coefficient was used as the dependent variable, and the most import ant
physicochemical characteristics of surface water selected on the basis of principal component
analysis for each use variant wereused as predictors)

| Variable | SAR ratio | Standard factor | Standard error | t | p |
|---|---|---|---|---|---|
| Surveypoints in the watercoursenext to arable land | | | | | |
| The constant of the equation | 9,46 | - | 5,35 | 0,23 | 3,50 |
| K$^+$ | -0,85 | 0,82 | 0,83 | -4,06 | 0,004 |
| pH | -2,50 | -0,39 | 5,34 | -2,25 | 0,32 |
| N-NO$_2$ | 0,009 | 0,05 | 0,02 | 0,36 | 0,78 |
| Ca$^{2+}$ | -0,83 | 8,14 | 1,67 | 3,97 | 0,15 |
| Mg$^{2+}$ | -0,37 | 0,04 | 0,01 | 0,01 | 0,04 |
| P-PO$_4^{3-}$ | -2,96 | 0,02 | 0,02 | -2,05 | 0,006 |
| Na$^+$ | 25,75 | 4,77 | 1,34 | -4,06 | 4,95 |
| Total suspended solids | 2,56 | 0,32 | 0,34 | 3,72 | 10,03 |
| Measurementpoints in the watercoursenext to grasslands | | | | | |
| The constant of the equation | 1,19 | - | 1,79 | 0,66 | 1,45 |
| Temperature of water | 0,04 | 0,01 | 0,42 | 0,10 | 0,51 |
| The constant of the equation | 0,12 | 0,02 | 1,27 | 0,09 | 0,92 |
| Fe$_{og.}$ | -0,02 | -0,45 | 0,006 | -4,00 | 0,01 |
| K$^+$ | 0,35 | 0,003 | 12,10 | 3,97 | 0,004 |
| Ca$^{2+}$ | -7,82 | -0,09 | 0,30 | 0,01 | 0,99 |
| Na$^+$ | -1,95 | -0,11 | 0,17 | -2,05 | 0,07 |
| Temperature of | -2,96 | -0,81 | 0,93 | -1,58 | 0,15 |



| | | | | | |
|---|---|---|---|---|---|
| water | | | | | |
| Measurementpoints in the watercoursenear the forests | | | | | |
| The constant of the equation | - | -0,90 | 0,24 | -3,49 | 0,01 |
| Temperature of water | -0,03 | -0,03 | 0,07 | -0,45 | 0,66 |
| ChZT-Mn | -0,11 | 0,10 | 0,15 | -0,72 | 0,49 |
| $Na^+$ | -0,04 | -0,04 | 0,29 | -0,51 | 0,88 |
| Conductance | -0,03 | 0,09 | 0,27 | 0,53 | 0,56 |
| $BZT_5$ | 0,008 | 0,009 | 0,22 | 0,03 | 0,97 |
| $Mg^{2+}$ | -0,20 | -0,13 | 0,09 | -2,12 | 0,07 |
| Dissolved oxygen | -0,46 | 0,003 | 0,004 | 0,21 | 0,99 |
| $Fe_{og.}$ | -0,03 | 0,19 | 0,06 | 0,15 | 0,24 |

## 4. Discussion

An innovative solution proposed in this dissertation is the combination of multidimensional statistics methods and a spatial model. During the preparation of the results, a close relationship was determined between selected physico-chemical characteristics of surface water quality using the spatial autoregression equation.

### 4.1. Influence of hydrodynamic parameters on the ionic composition of surface water

Determining the level of micro-pollutants, especially heavy metals, is essential for monitoring areas exposed to contamination in long-term studies. The ionic composition of surface water determines the intensity and direction of the chemical denudation process conditioned by lithology (Kim et al., 2012; Ziadat et al., 2015). It also determines the quality ofthe surface watercourses inventory and the hydrographic network's operation (Trivedi, 2010). Through PCA, the parent variables for each type of land usewere documented. For further spatial analysis, the physico-chemical features of surface water with the highest factor loadings were selected. A high negative correlation was only calculated in the first PCA factor for the concentration of $K^+$ and in the second PCA factor for concentrations of $Mg^{2+}$ and $Na^+$. This may be because these ions runoff from the catchment area.

Water self-purification processes are essential in the assessment and protection process (Moore and Langner, 2012; Halecki, 2015). Identifying the sources of nitrate pollution is one of the priority actions (Bu et al., 2017) because anthropogenic pollutants are different in surface water and may change seasonally (Xu et al., 2014; Hu et al., 2015). As for arable land, $K^+$





showed a negative relationship with the first factor of the PCA. A negative relationship was also
found for $Na^+$. This may mean that their concentration has been systematically decreasing. In
water samples collected near forested areas, a high factor load was calculated for the water
temperature and concentration of ChZT-Mn. They were also characterised by the highest
variance for the factor axes in all tested surface water samples and the highest factor loadings.
The water temperature could play a significant role in the concentration of ions dissolved in the
water of the flysch stream.

The state when the material was mixed in the water was considered in processing the

results. For this purpose, the transverse turbulence diffusion index was calculated to determine
transport intensity. The highest value of transport intensity, 1.02, was recorded in the riverbed of
the stream adjacent to arable land. According to PCA, the most important chemical parameter
associated with this land use typeis $Ca^{2+}$ and $P-PO_4^{3-}$. This is an interesting result because, when
assessing the impact of arable lands on the physico-chemical quality of surface water, particular
attention should be paid to the supply of phosphates and nitrates (Halecki et al., 2017). The
quantitative share of ions depends on the type of rocks and the physical properties of the
substrate, especially the infiltration coefficient, and for clastic weathering, on their mechanical
composition (Szostakiewicz-Hołownia, 2012). This aspect is of particular concernfor flysch
catchments built of the Magura and Sub-Magura layers (Starkel, 2006; Starkelet. al., 2007).

The diversified rock bottom relief impacts high water turbulence and,thus, the transport of

weathered loads (Priess et al., 2015; Szatten, 2016). Stationary tests with a high measurement
frequency are then performed to determine the relationship between flooding and suspension
concentration in particular years (Huan, 2011). On the other hand, in the Beskidy catchments,
which include the variability of the transported clastic weathered water, quantitative relationships
between hydrological parameters and their supply against the background of water erosion have
been shown (Brański, 1975; Brański, 1990). The most significant distance of full mixing was
calculated for points in surface water near arable land at 52.26 m.
**4.2. Relationships between hydrological parameters and their supply against the**
**background of water erosion**

The intensity of transport of chemical compounds depends on turbulence and dispersion

due to spatial differentiation of the flow velocity. Turbulent vortices create a locally



inhomogeneous and non-stationary velocity field, accelerating mixing (Loga, 2016). This
hydrodynamic parameter was calculated for each cross-section on each section of the stream to
approximate the transport mechanism of the investigated ion forms and other materials dissolved
in water. The Reynolds number reached the highest values in the stream bed adjacent to the
permanent grassland, similar to the tangential stress. The bottom sediment ranged from 32 to 41
mm in the entire catchment area. It was observed that material with more significant fractions is
collected at the flysch bed surrounded by permanent grasslands.
**4.3. Model of developed techniques of spatial autoregression**
Ions in water may be related to land use, and the detection of heavy metals in surface
water determines its economic value (Fu et al., 2014; Liao et al., 2016) and the health of the
inhabitants of the surrounding areas, especially in terms of therapeutic properties of Carpathian
rivers (Operacz et al., 2018). The spatial pattern and correlations between the variables in the
entire catchment area have been shown based on the spatial autoregression model, empirical
equations, and multivariate analysis. Furthermore, the research confirms that in a small flysch
catchment, it is possible to balance the material leached from the soil because the chemical
denudation process shows a relationship with the concentration of dissolved electrolytes in the
flysch stream (Halecki et al., 2019).
The model of developed techniques of spatial autoregression also enabled the inclusion of
spatial analysis in the extrapolation of hydrochemical data and the intensity of erosion processes
during multiple measurement series. The technique of spatial autoregression was chosen for
itsmore effective presentation of these relationships and was used to verify the analysed data in a
spatial system. The applied analysis helped to check whether the variables are affected by
different land use modifications. The independent variables were the physico-chemicalfeatures of
surface water at different land use variants. Spatial SAR model resultsshowed that $P-PO_4^{3-}$ and
$K^+$ ionsare leached mainlyfrom soil in arable lands.Moreover, a statistically significant
relationship between $K^+$ and $Fe_{og}$ cations was observed in surface water flowing from grasslands.
However, the SAR model of spatial autoregression did not detect statistically significant
differences inthe ionic composition of surface water flowing out of the forests. The land use
could have contributed to the differences generated by the model. Therefore, the methods used to


evaluate the surface washout and soil erosion studies on used slopes, especially for agricultural
purposes, should be improved (Bakker et al., 2008; Smolska et al., 2010).
The spatial autoregression model for the whole with the explanatory variable (the distance
of full mixing) detected the spatial dependencies for $P\text{-}PO_4^{3-}$, $Mn^{2+}$, $N\text{-}NO_2^-$, TDS, $Na^+$, $Mg^{2+}$,
$Ca^{2+}$ and $Mn^{2+}$. Since different hydraulic conditions were detected for each variant of land use, it
should be passed that the direction of the catchment development impacts the physico-chemical
quality of surface water.
**4.4. Recommendations for hydraulic tests and physico-chemical assessment of surface water**
Forecasting the physico-chemical quality of surface water and determining the
relationships between the concentration of metals, including heavy metals, requires the
determination of decisive indicators. Spatial autoregression is currently one of the most effective
methods of water quality forecasting for various sources of pollution (Yang et al., 2017). The
analysis of spatial autoregression primarily showed the spatio-temporal relationships at various
measuring points. Furthermore, including spatial analysis to determine many pollutants flowing
into the waters allows for solving technical problems related to surface washout and proper use of
arable land in the catchment area.
For spatial studies, the appropriate number of samples plays a crucialrole in building a
spatial model (Vallejos and Osorio 2014), especially in work related to the physico-chemical
quality of surface water (Mahjouri and Kerachian, 2011). This means that the assessment of
surface water quality features depends on the spatial scale. Therefore, the evaluation of surface
water quality should be combined with a spatial analysis (Dash et al., 2015). The amount of
leached total suspended solids delivered during water erosion is a value recorded during short-
term flooding rainfalls. So, based on the hydrochemical properties, it is possible to determine
surface water quality (Misaghi et al., 2017). Only relevant parameters should be considered in
assessing the physico-chemical quality of surface water in agricultural areas and grasslands. It is
vitalto choose a method consistent with the sample size (Griffith, 2005) in order to formulate a
dynamic model (Finley et al., 2012) or to calculate water quality indices (Yadav et al., 2015;
Naubi et al., 2016). Thus, the optimisation of spatial sampling is crucial (Hu and Wang, 2011) in
monitoring surface water pollution (Wu et al., 2005). The conclusions from this dissertation will
find practical application in isolating the primaryfactors favouring water erosion in streams with





larger catchment areas. In addition, recognising the effects of water erosion will be necessaryfor
evaluating the surface delivery of the material in further scientific research. The continuation of
scientific work in this field should focus on investigating the relationship between individual
washings and the concentration of material in the valleys of streams and rivers.
**5. Conclusions**
The effect of thiswork is important for determining the sources of diffuse pollution
flowing to the stream from agricultural areas in the periodic assessment of the physico-chemical
quality of surface water. The research thread undertaken in the dissertation will help determine
the intensity of water erosion and assess the sources of surface water supply. The assessment of
this phenomenon should aim at explaining the size of the load calculated into the delivery zones
of the transported material to the flysch stream and erosive factors, including susceptibility to
leaching. Some results will undoubtedly be used to create models of erosive feeding of alluvial
deposits from weathered flysch or surface wash, depending on the material delivery in the
catchment area. The results related to land use and its impact on the transport of chemical
compounds may be helpful in environmental management in mountain areas and chemical
monitoring in determining the course of erosion processes. Over the years, assessment of land use
changes may be of valuefor local residents involved in agriculture. Moreover, the assessment of
the short-term change in the concentration of the physico-chemical composition in the flysch
riverbed will be valuable for scientific purposes to develop a plan for the management and
elimination of pollutants resulting from anthropogenic pressure.
**Highlights:**
1. The physico-chemical composition of the flysch stream depends on the hydraulic
conditions.
2. The catchment use determines the intensity and direction of the chemical denudation
process.
3. The turbulent diffusion coefficient may influence the dissolved electrolytes in the
stream.
4. The spatial autoregression model detected relationships for nutrients leached from
arable land.



5. The assessment of the physico-chemical properties of surface water depends on the

spatial scale.


**Financial Support**

The publication was co-financed from the subsidy granted to the Krakow University of
Economics, Project nr 28/GGR/2021/POT.

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

Contributions to Probability and Statistics: Essays in Honor of Harold Hotelling. Stanford
University Press. s. 278–292, https://doi.org/10.2307/2285659, 1960.
Liao J., Wen Z., Ru X., Chen J., Wu H., Wei C. Distribution and migration of heavy metals in
soil and crops affected by acid mine drainage: public health implications in Guangdong Province,
China. Ecotoxicology and Environmental Safety 124: 460–469,
https://doi.org/10.1016/j.ecoenv.2015.11.023 , 2016
Loga M. Wody pod presją - praktyczny kurs oceny presji obiektów gospodarki komunalnej na
wody powierzchniowe praca zbiorowa. ISBN 978-83-937934-4-0. 2016.
MacCallum R. A comparison of factor analysis programs in SPSS, BMDP, and
SAS. Psychometrika, 48, 2: 223–231, https://doi.org/10.1007/BF02294017, 1983.
Mahjouri N., Kerachian R. Revising river water quality monitoring networks using discrete
entropy theory: the Jajrood River experience. Environmental Monitoring and Assessment 175(1-
4): 291–302, DOI:10.1007/s10661-010-1512-6, 2011.
Matta G., Srivastava S., Pandey R.R., Sain K.K., Assessment of physicochemical characteristics
of Ganga Canal water quality in Uttarakhand. Environment, Development and Sustainability 19,
2: 419–431, https://doi.org/10.1007/s10668-015-9735-x, 2017.
Mazur Z., Pałys S. Erozja wodna w zlewni lessowej na Lubelszczyźnie w latach 1956-1991.
AnnalesUniversitatisMariae Curie-Skłodowska–Sectio E 47, 219–229,
https://doi.org/10.12912/23920629/99168, 1992
Misaghi F., Delgosha F., Razzaghmanesh M., Myers B. Introducing a water quality index for
assessing water for irrigation purposes: A case study of the Ghezel Ozan River. Science of the
Total Environment 589: 107–116, https://doi.org/10.1016/j.scitotenv.2017.02.226, 2017.
Moore J.N., Langner H.W. Can a river heal itself? Natural attenuation of metal contamination in
river sediment. Environmental Science&Technology 46(5): 2616–2623,
https://doi.org/10.1021/es203810j, 2012.
Naubi I., Zardari N.H., Shirazi S.M., Ibrahim N.F., Baloo L. Effectiveness of water quality index
for monitoring Malaysian River water quality. Polish Journal of Environmental Studies 25(1):
231–239; DOI:10.15244/pjoes/60109; 2016.



Operacz A., Wąsik E., Hajduga M., Chmielowski K. Therapeutic Water in the Poprad Valley –
the Newest Development in the Polish Outer Carpathians. Polish Journal of Environmental
Studies 27(3): 1207–1217, https://doi.org/10.15244/pjoes/76036, 2018.
Oster J.D., Sposito G., Smith C.J. Accounting for potassium and magnesium in irrigation water
quality assessment. California Agriculture 70, 2: 71–76, doi: 10.3733/ca.v070n02p71, 2016.
Padmalal D., Remya S.I., Jyothi S.J., Baijulal B., Babu K.N., Baiju R.S., Water quality and
dissolved inorganic fluxes of N, P, SO$_4$, and K of a small catchment river in the Southwestern
Coast of India. Environmental Monitoring and Assessment 184, 3: 1541–1557,
DOI: 10.1007/s10661-011-2059-x, 2012.
Panagos P, Borrelli, P., Poesen, J., Ballabio, C., Lugato, E., Meusburger, K., Montanarella, L.,
Alewell, C. The new assessment of soil loss by water erosion in Europe. Environmental
Science & Policy 54: 438–44, https://doi.org/10.1016/j.envsci.2015.08.012, 2015.
Parmar D.L., Keshari A.K., Sensitivity analysis of water quality for Delhi stretch of the River
Yamuna, India. Environmental Monitoring and Assessment. 184, 3: 1487–1508,
https://doi.org/10.1007/s10661-011-2055-1, 2012.
Priess J.A., Schweitzer C., Batkhishig O., Koschitzki T., Wurbs D. Impacts of agricultural land-
use dynamics on erosion risks and options for land and water management in Northern Mongolia.
Environmental EarthSciences 73(2): 697–708, https://doi.org/10.1007/s12665-014-3380-9, 2015.
Rangel T.F.L.V.B., Diniz-Filho J.A.F., Bini, L.M. SAM: a comprehensive application for Spatial
Analysis in Macroecology. Ecography 33: 46–50, https://doi.org/10.1111/j.1600-
0587.2009.06299.x, 2010.
Saito M., Hayamizu K., Okada T. Temperature dependence of ion and water transport in
perfluorinated ionomer membranes for fuel cells. The Journal of Physical Chemistry B, 109(8):
3112–3119, https://doi.org/10.1021/jp045624w, 2005a.
Sapek A. Chlorki w wodzie na obszarach wiejskich (Chlorides in water from rural areas). Woda-
Środowisko-Obszary Wiejskie 8, 22: 263–281, ISSN: 1642-8145 2008.
Shapiro S.S., Wilk M.B. An analysis of variance test for normality (complete samples).
Biometrika 52: 591–611, https://doi.org/10.1093/biomet/52.3-4.591,1965.

708• Shi W., Xia J., Zhang X. Influences of anthropogenic activities and topography on water quality
in the highly regulated Huai River basin, China.
Environmental Science and Pollution Research 21: 21460–21474, DOI: 10.1007/s11356-016-
711 7368-8, 2016.

Shigut A.D., Liknew G., Irge D.D., Ahmad, T. Assessment of physico-chemical quality of
borehole and spring water sources supplied to Robe Town, Oromia region, Ethiopia, Applied
Water Science. 7, 1: 155–164, https://doi.org/10.1007/s13201-016-0502-4, 2017.
Smolska E. Spływ wody i erozja gleby na piaszczystym stoku w obszarze młodo glacjalnym –
pomiary poletkowe (Pojezierze Suwalskie, Polska NE). [W:] E. Smolska, J. Rodzik (red.),
Procesy erozyjne na stokach użytkowanych rolniczo (metody badań, dynamika i skutki). Prace i
Studia Geograficzne WGiSR UW 45: 197–214, ISSN: 0208-4589, 2010.





Smoroń S. Zagrożenia eutrofizacją wód powierzchniowych wyżyn lessowych małopolski
(Threats of eutrophication of surface waters of loess uplands of Lesser Poland). Woda-
Środowisko-Obszary Wiejskie. 12, 1, 37: 181−191, ISSN: 1642-8145, 2012.
Starkel L. Geomorphic hazards in the Polish Flysch Carpathians. Studia
GeomorphologicaCarpatho-Balcanica 40: 7–19, ISBN: 83-88549-56-1, 2006.
Starkel L. Złożoność czasowa i przestrzenna opadów ekstremalnych – ich efekty
geomorfologiczne i drogi przeciwdziałania im (Temporal and spatial complexity of extreme
rainfalls – their geomorphological effects and ways of counteract them). Landform Analysis 15:
65–80, ISSN:(p)1429-799X; (e)2081-5980, 2011.
Starkel L., Pietrzak M., Łajczak A. Wpływ zmian użytkowania ziemi i wzrostu częstotliwości
ekstremalnych opadów na obieg wody i erozję oraz ochronę zasobów przyrodniczych Karpat.
(The impact of changes in land use and the increase in the frequency of extreme rainfall on the
water cycle and erosion and protection of the natural resources of the Carpathians) Problemy
Zagospodarowania Ziem Górskich 54: 19–30, ISSN: 0137-5423 2007.
Stephens M.A. Tests based on edf statistics. Pp. 97–194 in D'Agostino, R.B. & Stephens M.A.
(eds.), Goodness-of-Fit Techniques. New York: Marcel Dekker, DOI:10.1201/9780203753064-4,
735 1986.

Szatten D. The estimation of suspended sediment transport using nefelometric and traditional
measurements of turbidity of water on an example of the cascade lower Brda River. Journal of
Education, Health and Sport 6(1): 64–72, https://doi.org/10.12775/PPS.2015.02.01.005, 2016.
Szostakiewicz-Hołownia M. Chemizm wód źródlanych zlewni Potoku Macelowego w Pieninach
(Chemism of spring waters of the Potok Macelowy catchment in the Pieniny Mountains). Pieniny
– Przyroda i Człowiek 12: 33–41, ISSN: 0033-2151, 2012.
Szüle B.. Introduction to data analysis. Publisher: Budapesti Corvinus Egyetem,
Közgazdaságtudományi Kar (Corvinus University of Budapest, Faculty of Economics). ISBN:
744 978-963-503 -619-6. 2016

Tabachnick BG, Fidell LS. Using Multivariate Statistics. Boston: Pearson Education Inc.,
ISBN:978-0-205-45938-4, 2007.
Tasdighi A., Arabi M., Osmond D.L. The relationship between land use and vulnerability to
nitrogen and phosphorus pollution in an urban watershed. Journal of Environmental Quality 46:
113–122, https://doi.org/10.2134/jeq2016.06.0239, 2017.
Teixeira, Z., Marques, J.C. Relating landscape to stream nitrate-N levels in a coastal eastern-
Atlantic watershed (Portugal). Ecological Indicators. 61: 693–706,
https://doi.org/10.1016/j.ecolind.2015.10.021, 2016.

753• Trivedi R.C. Water quality of Ganga River - An overview. Aquatic Ecosystem Health and
Management 13(4): 347–351, https://doi.org/10.1080/14634988.2010.528740, 2010.
Ulén B., von Brömsen C., Johansson G., Torstensson G., Stjernman-Forsberg L. Trends in
nutrient concentrations in drainage water from single fields under ordinary cultivation. Agric.
Ecosyst. Environ. 151: 61–69, https://doi.org/10.1016/j.agee.2012.02.005, 2012.
Vadde K.K., Wang J., Cao L., Yuan T., McCarthy A.J., Sekar R. Assessment of Water Quality
and Identification of Pollution Risk Locations in Tiaoxi River (Taihu Watershed), China. Water,
10 (2): 183, https://doi.org/10.3390/w10020183, 2018.



Vallejos R., Osorio F. Effective sample size of spatial process models. Spatial Statistics 9: 66–92,
https://doi.org/10.1016/j.spasta.2014.03.003, 2014.

763•   Vincent-Akpu I.F., Tyler A.N., Wilson C., Mackinnon G., Assessment of physico-chemical
properties and metal contents of water and sediments of Bodo Creek, Niger Delta, Nigeria.
Toxicological        & Environmental        Chemistry        97,        2,        135–144,
https://doi.org/10.1080/02772248.2015.1041526, 2015.
Wang R., Liu Z., Yao Z., Lei Y.. Modeling the risk of nitrate leaching and nitrate runoff loss
from intensive farmland in the Baiyangdian Basin of the North China Plain Environ Earth Sci 10,
72, 8: 3143–3157, https://doi.org/10.1007/s12665-014-3219-4, 2014
Weber E., Grattan S.R., Hanson B.R., Vivaldi G.A., Meyer R.D., Prichard T., Schwankl L.J.
Recycled water causes no salinity or toxicity issues in Napa vineyards. California Agriculture 68,
3, 59–67, DOI:10.3733/ca.v068n03p59, 2014.
Williams B., Brown, T., Onsman A. Exploratory factor analysis: A five-step guide for novices.
Australasian Journal of Paramedicine, 8, 3:1–13, https://doi.org/10.33151/ajp.8.3.93, 2010.
Wojtasik M., Szatten D. The balance of sediment supply by water erosion determined by USLE
model on the catchment of Brda river. Journal of Health Sciences 4, 11: 61–70, issn 1429-
9623/2300-665X. 2014.
778•   Wolman M.G., A method of samplingcoarseriver-bedmaterial. Transactions of the American
Geophysical Union EOS, 35 1954, pp. 951–956, DOI:10.1029/TR035I006P00951, 1954
Wu J., Zheng C., Chien C.C. Cost-effective sampling network design for contaminant plume
monitoring under general hydrogeological conditions. Journal of Contaminant Hydrology 77(1-
2): 41–65, https://doi.org/10.1016/j.jconhyd.2004.11.006, 2005.
Xu Y., Sun Q., Yi L., Yin X., Wang A., Li Y., Chen J. The source of natural and anthropogenic
heavy metals in the sediments of the Minjiang River estuary (SE China): implications for
historical      pollution.      Science      of      the      Total      Environment      493:      729–736,
https://doi.org/10.1016/j.scitotenv.2014.06.046, 2014.
Yadav K.K., Gupta N., Kumar V., Sharma S., Arya S. Water quality assessment of Pahuj River
using water quality index at Unnao Balaji, M.P., India. The International Research Journal of
Applied and Basic Sciences 19(1): 241–250, ISSN: 2251-838X, 2015.
Yang X., Liu Q., Luo X., Zheng Z. Spatial Regression and Prediction of Water Quality in a
Watershed    with    Complex    Pollution    Sources.    Scientific    Reports.    7:    8318,
https://doi.org/10.1038/s41598-017-08254-w, 2017.
Zhang Q., Chen H., Wu T., Jin T., Pan Z., Zheng J., Gao Y., Zhuang W. The opposite effects of
sodium and potassium cations on water dynamics. Chemical Science 8(2): 1429–1435,
10.1039/c6sc03320b, 2017.
Ziadat A.H., Jiries A., Berdanier B., Batarseh M. Biomonitoring of heavy metals in the vicinity of
copper mining site at Erdenet, Mongolia. Journal of Applied Sciences 15(11): 1297–1304,
DOI:10.3923/jas.2015.1297.1304, 2015.