# Peer review of "Spatial relationship between hydrodynamic and physico-chemical parameters of surface 1 2 water for a basin with shale rock series as an indicator of intensity and direction of 3 chemical denudation in the Western Carpathians Edyta Kruk1, Wiktor Halecki2, Marek Ryczek3</sup"

_Hydrology and Earth System Sciences, 2023_

## Referee Comment (RC1)

**Overall**, I think this manuscript would have benefitted from additional editing before submission. There are multiple typos and spelling mistakes. There are also references to the 'dissertation', equations are not numbered, several of the Tables not referred to in the text, misspelling of several references (e.g. Ulénet al., 2012). Clearly the authors have conducted a lot of work which is impressive, but the reporting here does not in my opinion do justice to this effort. Detailed comments are provide below, including itemised points for each section.

**Abstract/introduction** consists of a number of sentences stuck together to make paragraphs. Each sentence reflect a statement from a cited paper, but there is little or no linkage between the sentence. Thus, rather than reading a text where an argument is being constructed, I feel like I am reading a set of disparate sentences, with no guidance as to what is the main message. The end-result is that the introduction fails to effectively convey what is the scientific knowledge gap that is being addressed here? Is it methodological, or does it have to do with processes in a particular geographical area (as the title suggests)?

Abstract

- Multiple phrases needing clarification and/or grammar editing
- Physico-chemical what, per line 37
- Needs stronger conclusion
- Need to be consistent with parameters (K vs iron (Fe)); also, recommend using symbol for turbulent diffusion coefficient (this is a common, known parameter)

Introduction

- Include brief statement of intro on 'flysch' for non-specialist readers
- Lines 48-55: what about field-based monitoring approaches?
- Introduce where the Beskidy range is located, and its significance
- Improper spacing, erroneous phrasing/words and other grammatical errors common throughout manuscript. Some of these errors may be due to conversion to the pdf document (e.g., from Word), but the final submitted version should be carefully proof-read regardless
- Many references (e.g., line 72 using a ref for algae from 2012) are outdated; please refer to more current literature, especially for broader impact statements such as in the Introduction. Also, perhaps use a more global approach to studies if possible.
- Use parameters or acronyms consistently after initial definition; e.g., Na/sodium in line 75
- Line 87: reference which specific WHO standards, and include in reference list
- Line 89: no real need to emphasise both electrical conductivity and TDS as these are correlated?
- Section on DO starting line 91: this section is informative, but imbalanced on the importance of dissolved oxygen. The microbial component is emphasised well, as reflected by the BOD and COD discussion. However, the controlling (and more simple) parameter of direct DO concentration, and relevance to chemical parameters linked to high or low DO and corresponding environments (e.g., anoxia leading to release of dissolved chemical species, nutrients driving algal issues, fish kills, etc.) are not mentioned.

- Line 104: this reference and focus is misleading; you are not working with drinking water directly with this paper/study, or at least that has not yet been introduced. From an environmental perspective (which I think is the direction of this paper), dissolved oxygen and corresponding effects on the full range of chemical and biological parameters is key to water quality.
- Be more specific in intro: what basin, what are the useful values (line 117) and why? Emphasise what makes this study unique.

**Methods**

**Section 2.4**: This is a confusing section, which I think could be improved by a more careful description of the dataset on which the statistical tests are being applied, as well as a more careful description of the aim (what question(s) are you trying to answer here, and a rationale for why these particular tests were chosen in favour of others?

What is the purpose of the X2 test? Are you testing for a particular distribution?

Line 242: What 'data' are you referring to here?

Line 243-250: This is really hard to follow and does not give confidence that the authors really understand what is being done here?

It is not clear how much innovation the authors have brought to the statistical analysis framework. The spatial autoregressive model is presented with an reference to Rangel et al. (2010), yet there is also a reference later (in the discussion, line 470) to Yang et al. (2017) highlighting their contribution to spatial autoregressive models in water quality studies. Arguably, this should have been highlighted in the introduction or the method section.

- At what time of month were samples obtained?
- Very nice Figure 1; would also be good to include a broader-scale map highlighting where the region is within Poland
- Also very nice Figure 2; indicate in caption that slope is right panel, exposition is left panel (hard to read within figure legend)
- What is the catchment area? Include in text.
- Regarding sample volume, if you are using 1 dm3, I recommend stating this in the more common 1 litre unit.
- Line 142: I disagree with this statement as being definitive. Concentration may be correlated to flow, but not always, and not directly. As mentioned above, sediment conditions paired with DO concentration, algae presence, local geology all have strong control on concentration (regardless of flow)
- How were convection and flow measured?
- Please clarify/restate what is meant by 'punctual sediment in a cross-section' (line 147)
- Line 149: what is meant by an additional 250 measurements were taken? What analyses were performed, on these extra samples or the primary ones? More detail on sampling and analysis needed
- Equations need numbering (e.g., Equation 1 after line 169).

- Regarding first equation (after line 169), what does the parameter 'a' actually indicate per the slope of a straight line? Straight line related to the velocity measurements obtained in situ, or? As it reads, it seems a function of two constant parameters.
- For Reynolds number parameters, define each parameter (e.g., $Re_{max}$) to its definition in line 178
- Line 194: no need to redefine gravity (already defined above)
- Line 209: is the experimental area in the field? If so, please specify as it is not clear if these measurements were obtained in situ or in the collected water samples.
- Line 218: please confirm, AAS measured the reduced species of these metals? How was oxidation prevented?
- Line 222: The winkler test is standard methodology; no need to repeat/include it here. What is needed, however, is how you used this Winkler method to estimate BOD5. Was this performed over a period of 5 days (assuming so), and if so, how frequently were follow-up measurements obtained? How was the sample stored to prevent changes in DO, temperature, etc?
- Good detail on suspended solids analysis

 **Result section**: On the whole, I find the text hard to follow. I cannot map the presented results back to the theoretical testing framework. Results are presented without reference to the mathematical quantities defined in the method section. Thus, I am really struggling to understand what is being reported in the Tables. For example, in Table 7: what is a SAR ratio? What is the OLS ratio?

Line 357: How do you interpret the results in Table 7 to get to the conclusion that these chemicals are appropriate predictors? And what do you mean by appropriate?

Line 360: I can't find the p-values reported anywhere?

- Grassland sections of the stream are commonly referred to; however, this is not detailed on the map in Figure 1. Are grasslands what is also characterised as pasture, or where is the grassland part of the stream in Figure 1? Important to clearly define regions on map in Figure 1 using the same descriptive text as what is used throughout paper
- Tables 1 – 5: are these the annual averages, or averages of the monthly samples taken over the multi-year period or . .? There is a lot of data in these tables, and likely the result of a lot of effort, but not clear on specifically what it is.
- Following on from this point, the discussion in lines 313-323 goes through the data in these tables very quickly; very short, fast discussion without solid introduction to what the results are you will be discussion. Please expand this section to give further detail.
- Figure 3: please provide supporting detail in the text on what the scree plots are showing; this is not clear
- Table 6: refers to circadian concentrations of TDS; were daily measurements obtained? Recommend using 'daily' rather than circadian
- Line 360: refer to parameter for turbulent diffusive coefficient already defined: $D_{t,y}$
- Table 7: what is OLS ratio?

**Discussion**:

The first sentence in the introduction states that "An innovative solution proposed in this dissertation is the combination of multidimensional statistics methods and a spatial model." This is of course good, but was not defined as one of the aims of the article at the end of the introduction?

I would recommend removing the background discussions and justifications in the discussion. For example, remove lines 386-390, or incorporate them into the introduction if it is important for the study. This would help to present your discussion more clearly. Same for lines 386-390, and for the entire paragraph starting line 475.

Nice idea with highlights, even if I would just call them conclusions. But I would have liked to see them being more specific and perhaps also map onto the sub-section in Section 4 to make a clearer link between the discussion of the results and the conclusions.

Notable, I think the discussion is weak on highlighting limitations of the study, such as sample size, representing complex physical and chemical catchment processes using relatively simple linear models, and other factors which might have implications for your results.

- Line 380: should refer to this work as publication/study, not dissertation
- I struggle to follow the implications and key points in the discussion as I don't feel I was able to clearly follow the results.
- Link between water quality (physico-chemical parameters) and mixing (flow, turbulent diffusion, etc) unclear from the very detailed tables and statistical plots (I fear I got overwhelmed, but key results should be clearly presented)

**Conclusion**

- Extremely well written, clear conclusion
- The link to the title of this paper is unclear; it is mentioned once in line 57 but otherwise, no mention?